# Latrophilin GPCR signaling mediates synapse formation

**Richard Sando\*, Thomas C Südhof**

Department of Molecular & Cellular Physiology and Howard Hughes Medical Institute, Stanford University School of Medicine, Stanford, United States

**Abstract** Neural circuit assembly in the brain requires precise establishment of synaptic connections, but the mechanisms of synapse assembly remain incompletely understood. Latrophilins are postsynaptic adhesion-GPCRs that engage in trans-synaptic complexes with presynaptic teneurins and FLRTs. In mouse CA1-region neurons, Latrophilin-2 and Latrophilin-3 are essential for formation of entorhinal-cortex-derived and Schaffer-collateral-derived synapses, respectively. However, it is unknown whether latrophilins function as GPCRs in synapse formation. Here, we show that Latrophilin-2 and Latrophilin-3 exhibit constitutive GPCR activity that increases cAMP levels, which was blocked by a mutation interfering with G-protein and arrestin interactions of GPCRs. The same mutation impaired the ability of Latrophilin-2 and Latrophilin-3 to rescue the synapse-loss phenotype in Latrophilin-2 and Latrophilin-3 knockout neurons in vivo. Our results suggest that Latrophilin-2 and Latrophilin-3 require GPCR signaling in synapse formation, indicating that latrophilins promote synapse formation in the hippocampus by activating a classical GPCR-signaling pathway.

## Introduction

Although many proteins that shape synaptic functions have been identified, little is known about how synapses are generated. Few proteins are known to be essential for synapse formation in vertebrates or invertebrates. Increasing evidence supports the notion that bi-directional trans-synaptic signaling by adhesion molecules promotes synapse formation (*Sanes and Yamagata, 2009*; *Nusser, 2018*; *Ribic and Biederer, 2019*; *Araç and Li, 2019*; *Lie et al., 2018*; *Chubykin et al., 2007*; *de Wit and Ghosh, 2016*; *Südhof, 2018*). However, nearly all the candidate synaptic adhesion molecules that have been tested by rigorous genetic approaches are not required for synapse formation per se. Instead, some of these candidate synaptic adhesion molecules, such as SYG1 in *Caenorhabditis elegans* (*Shen et al., 2004*) or cadherins in the retina (*Duan et al., 2018*), direct neuronal processes to the correct position of future synapses. Other candidate synaptic adhesion molecules, such as neurexins and their ligands (reviewed in *Südhof, 2018*) or the SALM/Lrfn family (reviewed in *Lie et al., 2018*), specify the properties of synapses (*Missler et al., 2003*; *Nair et al., 2013*; *Varoqueaux et al., 2006*). A case in point are cerebellins, which connect presynaptic neurexins to postsynaptic GluD and DCC/neogenin adhesion molecules. Although deletion of cerebellin-1 or GluD2 causes robust parallel-fiber synapse loss in the cerebellum, synapses are only eliminated secondarily after they were established. Moreover, only a fraction of parallel-fiber synapses is lost, whereas all parallel-fiber synapses exhibit altered synaptic plasticity (reviewed in *Yuzaki, 2018*). In other parts of the brain, cerebellin-1 and −2 deletions also do not hinder initial synapse formation, but impair synapse function and also lead to secondary synapse elimination (*Rong et al., 2012*; *Seigneur and Südhof, 2018*; *Seigneur et al., 2018*). Only few adhesion molecules appear to be actually required for the initial establishment of synapses. Among these molecules are two families of adhesion-GPCRs, latrophilins and BAIs, that are essential for establishing synapses in all brain regions in which they have been tested (*Kakegawa et al., 2015*; *Sigoillot et al., 2015*;

\*For correspondence:
richard.sando@vanderbilt.edu

**Competing interests:** The authors declare that no competing interests exist.

*Anderson et al., 2017*; *Sando et al., 2019*; *Wang et al., 2020*). Studying the mechanism of action of these adhesion-GPCRs offers an avenue for understanding the basic processes underlying initial synapse formation.

Adhesion-GPCRs characteristically contain large extracellular sequences with multiple potential ligand-binding domains and a canonical GAIN domain that is coupled to the seven-transmembrane region module characteristic of GPCRs (reviewed in *Vizurraga et al., 2020*; *Morgan et al., 2019*; *Folts et al., 2019*; *Purcell and Hall, 2018*). Most adhesion-GPCRs additionally have a long cytoplasmic region. The three mammalian latrophilins (Lphn1-3) follow the same design principle as other adhesion-GPCRs. They feature N-terminal lectin- and olfactomedin-like ligand-binding domains, a serine-threonine rich and 'hormone binding' domain of unknown function, and a canonical GAIN domain (*Figure 1A*; *Sugita et al., 1998*). These domains are attached to a classical seven-transmembrane region GPCR module, followed by a long cytoplasmic region that interacts at the C-terminus with the PDZ-domain of SHANKs (*Kreienkamp et al., 2000*; *Tobaben et al., 2000*). The N-terminal lectin- and olfactomedin-like domains of latrophilins engage in transcellular interactions with multiple presynaptic ligands, including teneurins and FLRTs (*Sugita et al., 1998*; *O'Sullivan et al., 2012*; *Silva et al., 2011*; *Boucard et al., 2014*). Multiple latrophilin isoforms are generally co-expressed in the same neurons. CA1 pyramidal neurons of the hippocampus co-express Lphn2 and Lphn3 that are targeted to distinct dendritic domains of the same neuron and are essential for formation of different excitatory synaptic inputs (*Anderson et al., 2017*; *Sando et al., 2019*). Moreover, Lphn2 and Lphn3 are redundantly required in the cerebellum for parallel-fiber synapse formation (*Zhang et al., 2020*). The function of postsynaptic Lphn3 in hippocampal synapse formation requires intact binding sites for both presynaptic teneurins and FLRTs, indicating that latrophilins may mediate synapse specificity by acting as coincidence detectors that respond to two input signals (*Sando et al., 2019*).

The central role of latrophilins as postsynaptic adhesion-GPCRs in synapse formation raises the question whether latrophilins actually function as GPCRs in synapse formation. An intrinsic GPCR signaling activity was suggested for several adhesion-GPCRs, including latrophilins (*Purcell and Hall, 2018*; *Nazarko et al., 2018*; *Müller et al., 2015*; *Ovando-Zambrano et al., 2019*; *Scholz et al., 2017*; *Araç et al., 2016*; *Kishore et al., 2016*; *Liebscher et al., 2014*; *Vizurraga et al., 2020*). However, few data exist for the physiological role of mammalian adhesion-GPCRs in general, and for their GPCR activity in particular. Whether latrophilins physiologically act as GPCRs remains unclear. Given the ubiquity of GPCRs in all cells, especially in neurons, and the relative non-specificity of GPCR signaling, a GPCR-signaling mechanism in synapse formation would be surprising and imply a tight spatial segregation of such signals in neurons (*Ellisdon and Halls, 2016*; *Patriarchi et al., 2018*). Indeed, other potential mechanisms for the function of latrophilins in synapse formation are equally plausible, for example a pure adhesion mechanism. To address this question, we here examined the GPCR signaling functions of latrophilins in cultured neurons and in vivo, testing this question using multiple independent approaches for two different latrophilins to ensure validity. Our data show that latrophilin-dependent synapse formation requires their GPCR-signaling activity that may have been adapted to synapses as a unique intercellular junction by use of a high degree of spatial compartmentalization.

## Results

### Design and validation of Lphn2 and Lphn3 mutations

To examine whether latrophilins act as GPCRs in synapse formation, we generated three types of mutants (*Figure 1A*). The Lphn3-ECD mutant deletes all transmembrane regions and cytoplasmic sequences of Lphn3 and attaches its extracellular domains to the membrane using a GPI-anchor, thereby preserving only the Lphn3 adhesion function and deleting all of its potential signaling capacity (*Gokce and Südhof, 2013*). The Lphn2-T4L and Lphn3-T4L mutants contain an insertion of a T4 lysozyme (T4L) sequence into the 3$^{rd}$ intracellular loop of Lphn2 or Lphn3, which abolishes G-protein coupling and impairs arrestin signaling to these GPCRs (*Rosenbaum et al., 2007*; *Cherezov et al., 2007*; *Thorsen et al., 2014*). The Lphn3-ΔCt mutant truncates most of the cytoplasmic C-terminal sequences, thereby ablating cytoplasmic interactions (*Figure 1A*). Of these three types of mutants, the Lphn3-ECD mutant asks whether Lphn3 might simply act as an adhesion molecule, the Lphn2-T4L and Lphn3-T4L mutants test whether GPCR-signaling is involved in latrophilin

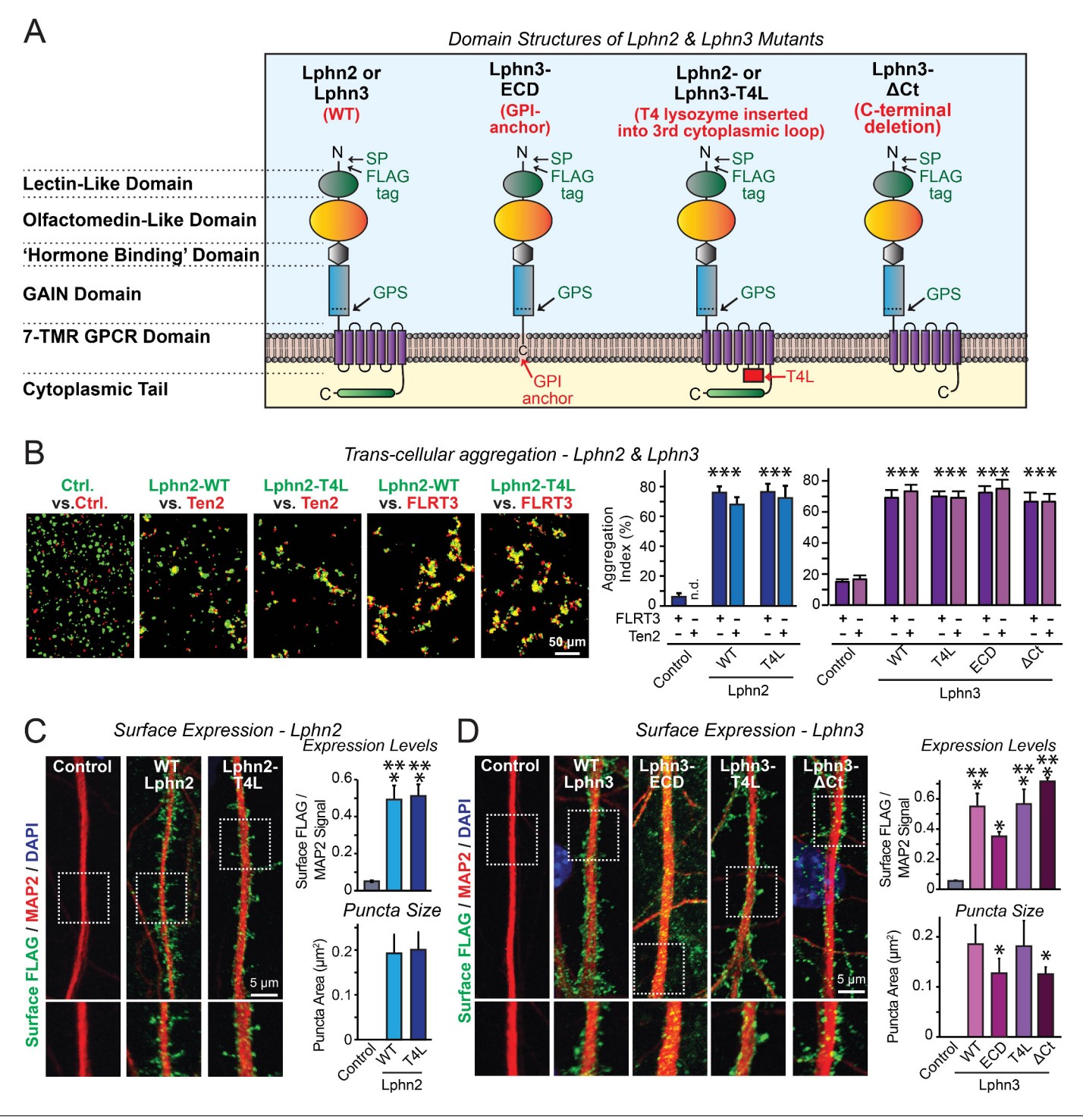

**Figure 1.** Latrophilin-2 (Lphn2) and Latrophilin-3 (Lphn3) signal transduction mutants engage in teneurin- and FLRT-mediated intercellular interactions on the cell surface. (A) Domain structures of wild-type and mutant Lphn2 and Lphn3. Mutants contain either only the extracellular Lphn3 domains attached to the membrane via a GPI-anchor (Lphn3-ECD), block G-protein coupling (Lphn2-T4L and Lphn3-T4L), or truncate the intracellular C-terminal tail of Lphn (Lphn3-ΔCt). (B) Wild-type and mutant Lphn2 and Lphn3 proteins are robustly expressed on the cell surface of HEK293 cells and efficiently mediate trans-cellular adhesion by binding to FLRT3 and Ten2 ligands (*left*, representative images of transcellular aggregation assays as indicated (Ctrl = empty plasmid); *right*, quantification of the aggregation efficiency for Lphn2 and Lphn3 constructs). (C and D) Wild-type and mutant Lphn2 (C) and Lphn3 proteins (D) are robustly expressed on the cell surface of hippocampal neurons (*left*, representative images of dendritic segments of surface-labeled cultured hippocampal neurons infected with lentiviruses expressing the indicated proteins); *right*, summary graphs of the surface levels of

*Figure 1 continued on next page*

*Figure 1 continued*

FLAG-tagged Lphn3 forms relative to MAP2 signal, as well as the FLAG puncta area. Numerical data are means ± SEM. Statistical significance was assessed by two-tailed t-test (For **C**, puncta area) or one-way ANOVA with post-hoc Tukey tests for multiple comparisons (For **B**, **C** and **D**) (*** denotes $p<0.001$; * denotes $p<0.05$). For additional images and control experiments, see *Figure 1—figure supplement 1*.

The online version of this article includes the following source data and figure supplement(s) for figure 1:

**Source data 1.** Latrophilin-2 (Lphn2) and Latrophilin-3 (Lphn3) signal transduction mutants engage in teneurin- and FLRT-mediated intercellular interactions on the cell surface.

**Figure supplement 1.** Representative images of aggregation assays for wild-type and mutant Lphn3 constructs (**A**) and cell-surface labeling control (**B**).

function, and the Lphn3-ΔCt mutant examines the possible role of binding partners for the cytoplasmic tail, such as Shanks (*Kreienkamp et al., 2000*; *Tobaben et al., 2000*). All wild-type (WT) and mutant Lphn2 and Lphn3 proteins were encoded with an N-terminal FLAG epitope tag to enable their detection on the cell surface (*Figure 1A*).

We first investigated whether the Lphn2 and Lphn3 mutants are transported properly to the cell surface of transfected HEK293 cells, and whether they are competent to engage in the normal trans-cellular adhesion interactions of Lphn2 and Lphn3 with FLRTs and teneurins. To test this question, we measured the ability of the mutants to support cell-aggregation mediated by these interactions. We generated two populations of HEK293 cells that co-expressed either FLRT3 or teneurin-2 (Ten2) with EGFP, or WT or mutant Lphn2 or Lphn3 with tdTomato, using transfections with empty plasmid DNA as a control. We then mixed the two cell populations of HEK293 cells and measured cell aggregation (*Boucard et al., 2014*). The mutations had no effect on the surface transport of Lphn2 or Lphn3, nor did they impair the ability of Lphn2 or Lphn3 to form trans-cellular adhesion complexes with FLRT3 or Ten2, suggesting that the mutant latrophilins are folded and functional in protein-protein interactions (*Figure 1B*; *Figure 1—figure supplement 1*).

We next expressed the WT and mutant Lphn2 and Lphn3 proteins in cultured hippocampal neurons by lentiviral delivery. Using immunocytochemistry for the surface FLAG-epitope present in all latrophilin proteins and for the dendritic marker MAP2, we visualized the surface transport and sub-cellular localization of WT and mutant Lphn2 and Lphn3 proteins in the neurons (*Figure 1C and D*). All Lphn2 and Lphn3 proteins were localized to the cell surface of neurons, with no apparent difference in localization between WT and mutant proteins. Quantifications documented that all mutant proteins were as well expressed on the surface as WT proteins (*Figure 1C and D*). Only the levels of the GPI-anchored Lphn3-ECD protein were slightly lower on the cell surface than those of the other proteins (*Figure 1C and D*). We performed an additional control experiment to test if the fixation procedure used permeabilizes cells in the absence of detergent. Primary hippocampal cultures were infected with lentiviral FLAG-Lphn3 WT vs. uninfected controls at 3DIV and labeled for FLAG together with the intracellular excitatory postsynaptic scaffold protein Homer1 at 12DIV. All conditions were subjected to the same fixation procedure utilized in our cell-surface labeling experiments (4% PFA/4% sucrose/PBS for 20 min at 4°C) followed by either permeabilization with 0.2% Triton X-100/PBS for 5 min or non-permeabilization. Permeabilization with 0.2% Triton X-100/PBS exposed robust Homer1 signal, suggesting that our fixation procedure alone generates minor membrane permeability (*Figure 1—figure supplement 1B*). This control supports that our assay is sampling predominately surface FLAG-tagged protein in non-permeabilized conditions (*Figure 1—figure supplement 1B*). Viewed together, these results show that the various Lphn2 and Lphn3 mutations do not impair their surface transport and extracellular ligand interactions.

## Lphn2 and Lphn3 are constitutively active GPCRs whose signaling is blocked by the T4L mutation

To test whether latrophilins exhibit GPCR activity and how such activity is altered by the Lphn2 and Lphn3 T4L mutations, we examined whether latrophilins alter cellular cAMP levels. We co-transfected HEK293T cells with the fluorescent cAMP indicator ('pink Flamindo'; *Harada et al., 2017*), EGFP as an internal fluorescence standard, Gαs (a G-protein) together with Gβ1γ2, and various WT and mutant Lphn2 and Lphn3 constructs. In addition, we co-transfected subsets of cells with the high affinity cAMP-specific phosphodiesterase PDE7b as a negative control (*Hetman et al., 2000*). Overexpressed PDE7b should eliminate all cytoplasmic cAMP, enabling us to estimate the

background fluorescence signal. Imaging revealed that without latrophilins, PDE7b had no effect on pink Flamindo fluorescence, suggesting that the observed signal represents background and that the basal levels of cAMP in HEK293T cells are undetectably low at these acquisition settings (*Figure 2A*; *Figure 2—figure supplement 1*). Expression of WT Lphn2 and Lphn3 caused a constitutive elevation of pink Flamindo fluorescence. This fluorescence increase was blocked by PDE7b, demonstrating that the pink Flamindo signal is specific for cAMP and that Lphn2 and Lphn3 increase cAMP levels (*Figure 2A*). The T4L mutation completely abolished the specific pink Flamindo signal, suggesting that it blocks the GPCR activity of Lphn2 and Lphn3 (*Figure 2A*).

A potential problem of the cAMP measurements with pink Flamindo is the rather high background signal of pink Flamindo that is cAMP-independent (*Figure 2A*). To further ensure specificity of the measurements, we designed an indicator construct for measuring the activity of cAMP-dependent protein kinase A (PKA). In this construct, a PKA-phosphorylation sequence for which a phospho-specific-antibody is available is fused N-terminally to tdTomato, which functions as an internal fluorescent control (*Figure 2B*). Immunocytochemical analysis of the PKA-indicator protein in HEK293T cells as a function of stimulation with forskolin and IBMX revealed a strong cAMP-dependent signal with a much better signal-to-noise ratio than the fluorescent pink Flamindo indicator (*Figure 2C*).

We then used the newly constructed PKA-indicator to examine the effect of WT and mutant Lphn2 and Lphn3 proteins on PKA activity. Co-expression of the PKA indicator with Gαs and various Lphn2 and Lphn3 proteins, without or with PDE7b, confirmed that Lphn2 and Lphn3 both activate cAMP signaling (*Figure 2D*). The Lphn2- and Lphn3-induced PKA-indicator signal was greatly attenuated by PDE7b, confirming that it is due to cAMP. The T4L-mutations of Lphn2 and Lphn3 similarly suppressed the PKA-indicator signal induced by Lphn2 or Lphn3, and PDE7b had no additional effect (*Figure 2D*). Together, these data suggest that Lphn2 and Lphn3 stimulate cAMP production in cells in a manner dependent on a GPCR transduction mechanism.

## Lphn3 function in cultured hippocampal neurons requires both its GPCR activity and its cytoplasmic sequence

We infected hippocampal neurons cultured from Lphn3 conditional KO mice with lentiviruses encoding ΔCre (control) or Cre, without or with addition of Lphn3 rescue constructs. We measured synapse density by immunolabeling the neurons for vGluT1 and PSD-95, which are pre- and postsynaptic markers of excitatory synapses, respectively (*Figure 3A*). As shown previously (*Sando et al., 2019*), the Lphn3 deletion robustly decreased (~40%) the excitatory synapse density in cultured hippocampal neurons. This decrease was fully rescued by WT Lphn3 but not by any of the three Lphn3 mutants (Lphn3-ECD, Lphn3-T4L, and Lphn3-ΔCt) (*Figure 3A*). This result suggests that Lphn3 functions as a GPCR in synapse formation and that this function depends additionally on the cytoplasmic sequences of Lphn3.

To confirm this conclusion, we performed whole-cell patch-clamp recordings of mEPSCs (*Figure 3B*). The Lphn3 deletion caused a ~40% decrease in mEPSC frequency, which corroborates the decrease in synapse density, but had only minor effects on the mEPSC amplitude (*Figure 3C and D*). Again, this phenotype was fully rescued by WT Lphn3 but not by the three Lphn3 mutants tested, which were inactive even though they were fully transported to the surface of dendritic spines (*Figures 1*, *3C and D*). These results suggest that Lphn3 functions as a GPCR in synapse formation in cultured neurons, and that this function additionally requires their long cytoplasmic tail region.

## Lphn2 and Lphn3 function as GPCRs in the hippocampus in vivo

Do Lphn3 – and by extension, Lphn2 – function as GPCRs in vivo? To address this question, we sparsely infected pyramidal neurons in the CA1 region of the hippocampus of newborn Lphn2 or Lphn3 conditional KO mice with lentiviruses encoding Cre, without or with co-expression of WT or G-protein-binding deficient Lphn2 (Lphn2-T4L) or Lphn3 (Lphn3-T4L). Three weeks later, we measured synaptic connectivity via Schaffer-collateral and entorhinal inputs in infected CA1 pyramidal neurons using whole-cell patch-clamp recordings in acute slices (*Figures 4* and *5*). We monitored AMPA-receptor-mediated synaptic responses in input-output curves to control for differences in stimulus strength, using a dual stimulation approach that enables separate measurements of Schaffer

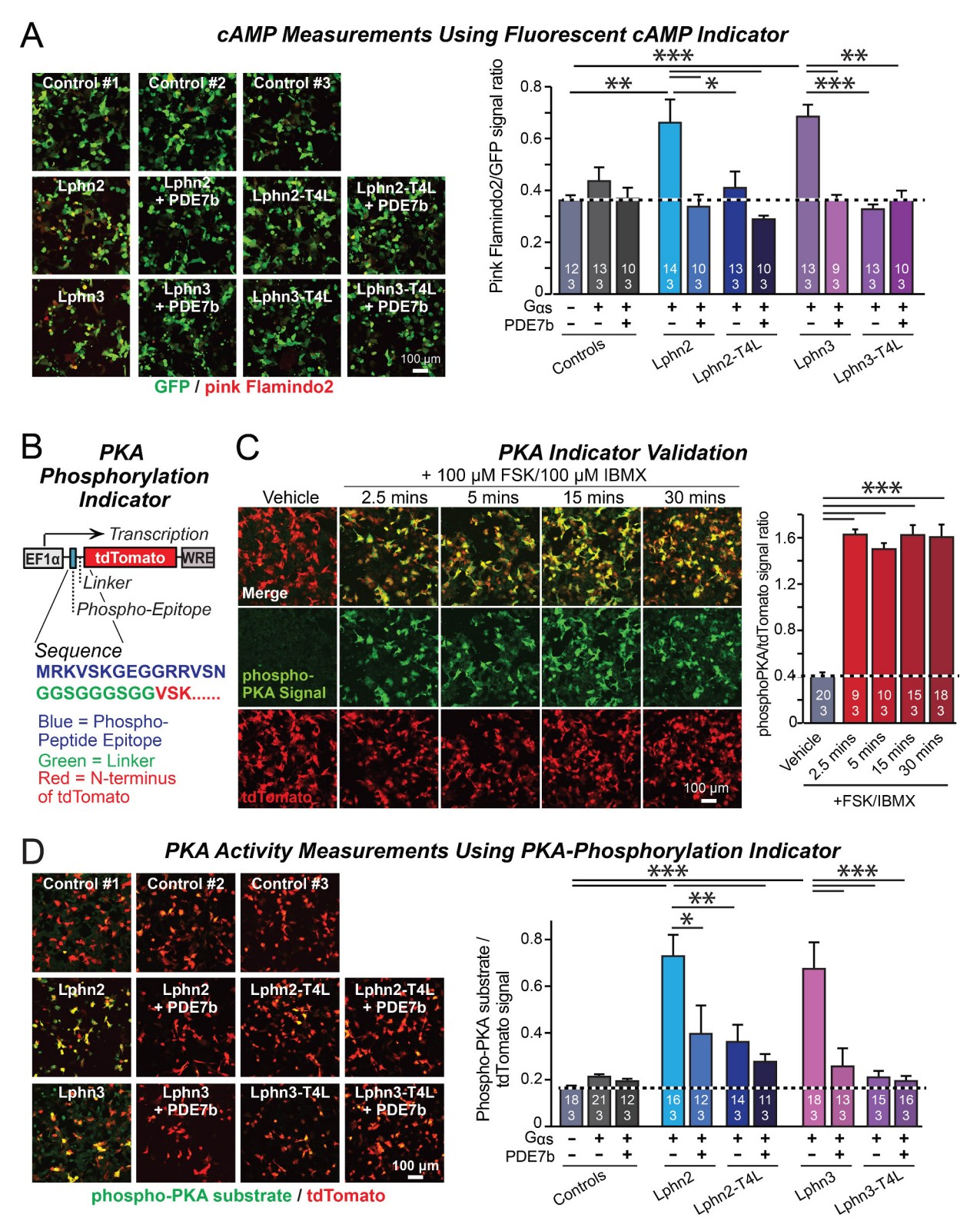

**Figure 2.** Lphn2- and Lphn3-induced cAMP signaling is blocked by insertion of T4-lysozyme into the third cytoplasmic loop. (**A**) Expression of wild-type Lphn2 or Lphn3 in HEK293T cells together with Gαs and Gβγ constitutively induces cAMP accumulation. The T4L mutation and co-expression of the cAMP-specific phosphodiesterase PDE7b prevent cAMP increases by Lphn2 or Lphn3. The Pink Flamindo cAMP reporter (***Odaka et al., 2014***; ***Harada et al., 2017***) was used to measure cAMP levels in live cells, and was quantified relative to the GFP signal as an internal standard. (**B**) Design of

*Figure 2 continued on next page*

*Figure 2 continued*

a new phospho-PKA indicator to measure PKA activation (*top*, plasmid design; *bottom*, sequences of the N-terminal PKA phospho-peptide substrate (blue) that was fused to the N-terminus of tdTomato (red) via a glycine-rich linker (green)). (**C**) Characterization of the phospho-PKA indicator in HEK293T cells. Treatment of cells with 100 μM Forskolin (FSK)/100 μM IBMX induces rapid phosphorylation of the phospho-PKA substrate (*left*, representative images; *right*, quantification of the immunocytochemical phospho-PKA signal relative to the tdTomato signal as an internal standard as a function of FSK/IBMX). (**D**) Lphn2 and Lphn3 constitutively activate PKA signaling in a manner blocked by the T4L-mutation. Experiments were performed as in A, but with the PKA-indicator described in B and C (*left*, representative images; *right*, summary graph). Numerical data are means ± SEM (number of analyzed ROIs/experiments in bars). Statistical significance was assessed by one-way ANOVA with post-hoc Tukey tests for multiple comparisons (For **A**, **C** and **D**) (*** denotes p<0.001; ** denotes p<0.01; * denotes p<0.05). For more representative images, see *Figure 2—figure supplement 1*.

The online version of this article includes the following source data and figure supplement(s) for figure 2:

**Source data 1.** Lphn2- and Lphn3-induced cAMP signaling is blocked by insertion of T4-lysozyme into the third cytoplasmic loop.

**Figure supplement 1.** Representative images of cAMP measurements using Pink Flamindo (**A**) and of PKA activity measurements using a new reporter protein (**B**) in HEK293T cells expressing wild-type and mutant Lphn2 and Lphn3 constructs.

collateral- and entorhinal cortex-derived synapses (*Anderson et al., 2017*; *Sando et al., 2019*). As a control, we measured uninfected neurons in acute slices from the contralateral side of the same mice.

The Lphn2 deletion selectively impaired entorhinal cortical synaptic inputs but not Schaffer-collateral synaptic inputs, confirming previous conclusions (*Anderson et al., 2017*). This phenotype was rescued by WT but not by T4L-mutant Lphn2 that is unable to mediate GPCR signaling (*Figure 4*). The Lphn3 deletion, conversely, selectively impaired Schaffer collateral- but not entorhinal cortex-derived synaptic inputs, again confirming previous findings (*Sando et al., 2019*). This phenotype was also rescued by WT but not by T4L-mutant Lphn3 (*Figure 5*), confirming that GPCR signaling is essential for latrophilin function in synapse formation.

The fact that the input-output curves in the electrophysiological recordings (*Figures 4* and *5*) are superimposable in these experiments attests to the solidity of the findings and the validity of the approach for measuring synaptic connectivity. However, a decrease in synaptic strength might be due to other causes than a loss of synaptic connectivity. To independently test these results, we used retrograde tracing of synaptic connections with pseudo-rabies viruses (*Callaway, 2008*; *Figure 6A*). Quantitative retrograde monosynaptic rabies tracing allows in vivo analyses of synaptic connections formed by a specific postsynaptic neuron (*Sando et al., 2019*). We employed a sparse rabies virus tracing strategy to assess the abundance of synaptic inputs onto hippocampal CA1 neurons of Lphn3 cKO mice expressing Cre without or with co-expression of WT Lphn3 or T4L Lphn3. Postnatal day 0 (P0) WT or Lphn3 cKO mice were sparsely infected monolaterally with low titer (1 × 10$^7$ IU/mL) lentiviruses expressing Cre or Cre together with WT Lphn3 or T4L Lphn3. The same CA1 region was subsequently infected with Cre-inducible rabies complementing AAVs necessary for rabies infection (AAV2/5 CAG-FLEX-TCB-mCherry and AAV2/8 CAG-FLEX-RG) at P21. The same CA1 region was subsequently infected with pseudotyped rabies RbV-CVS-N2c-deltaG-GFP (EnvA) at P35. Mice were perfused 5 days after rabies infection to diminish the possibility of rabies toxicity, sectioned, and counter-stained with DAPI. Images of the CA1, ipsilateral/contralateral CA3 and ipsilateral entorhinal cortex were collected with a 20X objective and the ratio of infected cells/DAPI quantified. Again, the Lphn3 conditional KO impaired synaptic connectivity (*Figure 6B and C*). This phenotype of rescued by expression of WT Lphn3, but not by the T4L-mutant Lphn3 unable to mediate GPCR signaling (*Figure 6B and C*). Viewed together, these data indicate that latrophilins function as postsynaptic GPCRs that control input-specific excitatory synapse formation in the hippocampus via a GPCR-dependent mechanism.

## Discussion

Here, we demonstrate that Lphn2 and Lphn3 are constitutively active GPCRs that enhance cAMP levels when expressed in HEK293 cells, and that the GPCR activity of Lphn2 and Lphn3 is essential for their ability to promote synapse formation in vivo. In addition, we show that Lphn3 requires its cytoplasmic sequences for supporting synapse formation. Based on these findings, we propose that latrophilins act in synapse formation by transmitting a GPCR-mediated signal, possibly by elevating

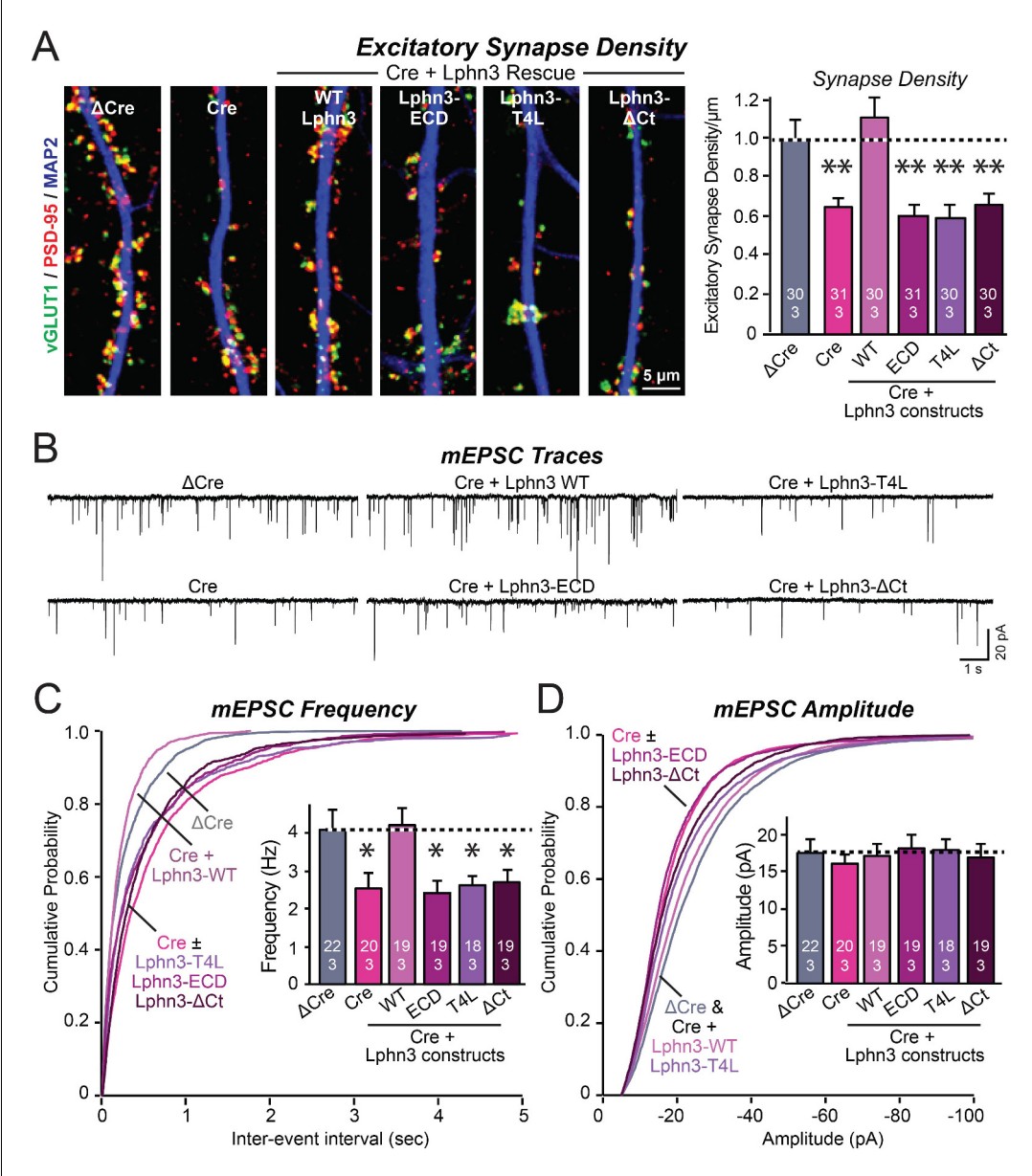

**Figure 3.** Mutations that disrupt Lphn3 signal transduction abolish the ability of Lphn3 to rescue the synapse formation in Lphn3-deficient cultured hippocampal neurons. (**A**) Mutations of Lphn3 that delete its transmembrane regions (Lphn3-ECD), block GPCR signal transduction (Lphn3-T4L), or delete its cytoplasmic tail (Lphn3-ΔCt) abolish the ability of Lphn3 to rescue the decreased excitatory synapse density of Lphn3-deficient neurons despite engaging in surface-ligand interactions. Hippocampal neurons cultured from Lphn3 cKO mice were infected at DIV3 with lentiviruses encoding ΔCre (control) or Cre without or with co-expression of the indicated Lphn3 rescue proteins, and were analyzed at DIV14-16. (**B–D**) The same mutations as analyzed in A also abolish the ability of Lphn3 to rescue the decreased mEPSC frequency of Lphn3-deficient cultured hippocampal neurons. Data are means ± SEM (numbers of analyzed cells/experiments are indicated in bars). Statistical significance was assessed by one-way ANOVA with post hoc Tukey tests for multiple comparisons (For **A**, **C** and **D**) (** denotes p<0.01; * denotes p<0.05). See *Figure 3—figure supplement 1* for additional expression controls and image data.

The online version of this article includes the following source data and figure supplement(s) for figure 3:

**Source data 1.** Mutations that disrupt Lphn3 signal transduction abolish the ability of Lphn3 to rescue the synapse formation in Lphn3-deficient cultured hippocampal neurons.

**Figure supplement 1.** Immunoblotting analyses of neurons expressing Lphn2 (**A**) and Lphn3 constructs (**B**), and demonstration that expression of various Lphn3 constructs does not grossly alter synapse morphology (**C**).

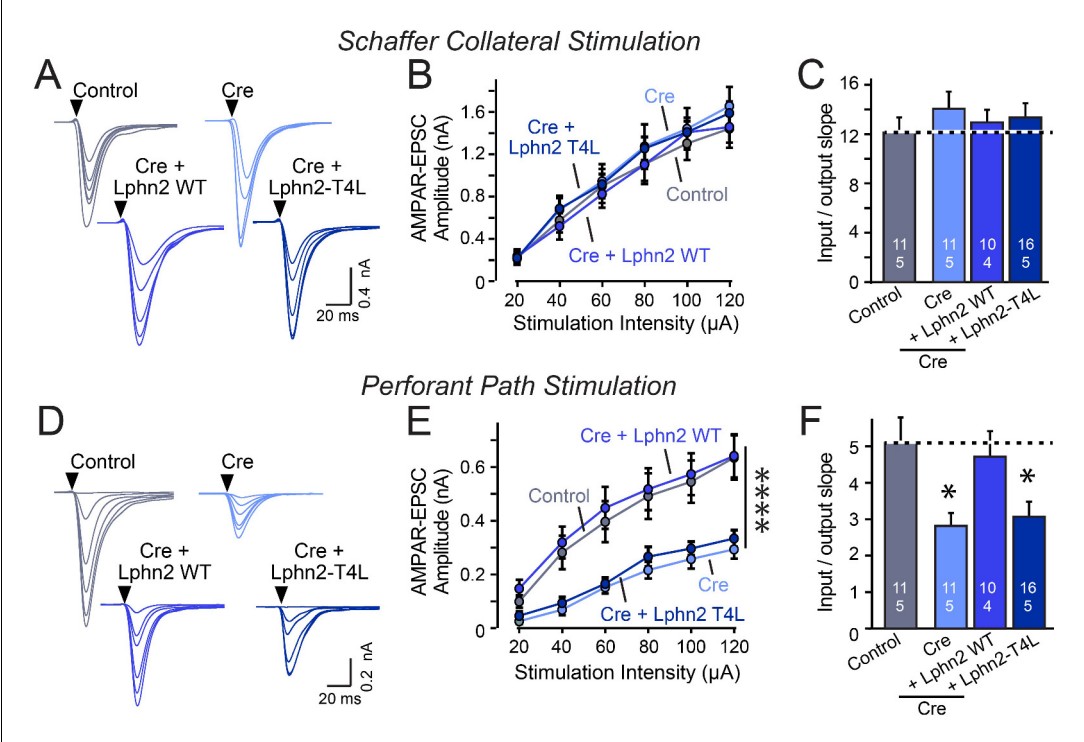

**Figure 4.** Abolishing GPCR signaling of Lphn2 by mutation of its third cytoplasmic loop blocks rescue of perforant-path synaptic connectivity in Lphn2-deficient CA1 region neurons in vivo. Data are from patch-clamp whole-cell recordings in acute hippocampal slice from Lphn2 conditional KO mice at P21-25. The CA1 region of the mice was injected at P0 with low titers of lentiviruses expressing either Cre alone, or Cre together with the indicated rescue constructs to induce sparse infection of CA1 pyramidal neurons (*Sando et al., 2019*). Control recordings were from uninfected cells in slices in the opposite uninfected hemisphere. (**A–C**) Sparse conditional KO of Lphn2 in CA1-region neurons has no effect on the synaptic responses mediated by Schaffer-collateral inputs, and overexpression of wild-type or T4L-mutant Lphn2 does not alter these responses (A, sample traces; B, input-output curves; C, summary graph of the slopes of the input-output curves). (**D–F**) Sparse conditional KO of Lphn2 in CA1-region neurons impairs synaptic responses mediated by entorhinal cortex inputs in a manner that can be rescued by wild-type but not T4L-mutant Lphn2 (D, sample traces; E, input-output curves; F, summary graph of the slopes of the input-output curves). Data are means ± SEM (numbers of cells/mice are indicated in bars). Statistical significance was assessed by one-way (For **C** and **F**) or two-way ANOVA with post hoc Tukey tests for multiple comparisons (For **B** and **E**) (**** denotes p<0.0001; * denotes p<0.05). See *Figure 4—figure supplement 1* for cellular capacitance and membrane resistance measurements. The online version of this article includes the following source data and figure supplement(s) for figure 4:

**Source data 1.** Abolishing GPCR signaling of Lphn2 by mutation of its third cytoplasmic loop blocks rescue of perforant-path synaptic connectivity in Lphn2-deficient CA1 region neurons in vivo.

**Figure supplement 1.** Intrinsic electrical properties of CA1 neurons are not significantly altered by the Lphn2 and Lphn3 deletions and by wild-type and T4L-mutant Lphn2 and Lphn3 rescues in the CA1 region of the hippocampus in vivo.

**Figure supplement 1—source data 1.** Intrinsic electrical properties of CA1 neurons in Lphn2 and Lphn3 rescue experiments.

cAMP. This is a surprising finding given that GPCRs are present in many neuronal subcompartments where they mediate multifarious regulatory functions (*Patriarchi et al., 2018*; *Sanderson et al., 2018*; *Scott et al., 2013*; *Torres-Quesada et al., 2017*). Thus, the latrophilin GPCR signal most likely is translated into specific organizational activities during synapse formation in a context-dependent manner, either via interactions of latrophilins with postsynaptic proteins such as SHANKs that are mediated by their cytoplasmic sequences, or via parallel signals induced by other trans-synaptic adhesion complexes.

Three lines of evidence support our conclusions. First, analysis of WT and mutant Lphn2 and Lphn3 in HEK293 cells demonstrated that WT Lphn2 and Lphn3 exhibit constitutively active GPCR activity that elevates cAMP levels, and that this activity is blocked by the T4L mutation, which is known to abolish GPCR signaling. Second, analysis of cultured neurons demonstrated that the Lphn2 and Lphn3 mutations do not impair surface transport of Lphn2 and Lphn3 or targeting to dendritic spines, but that the Lphn3 mutations abolish its ability to rescue the decrease in synapse

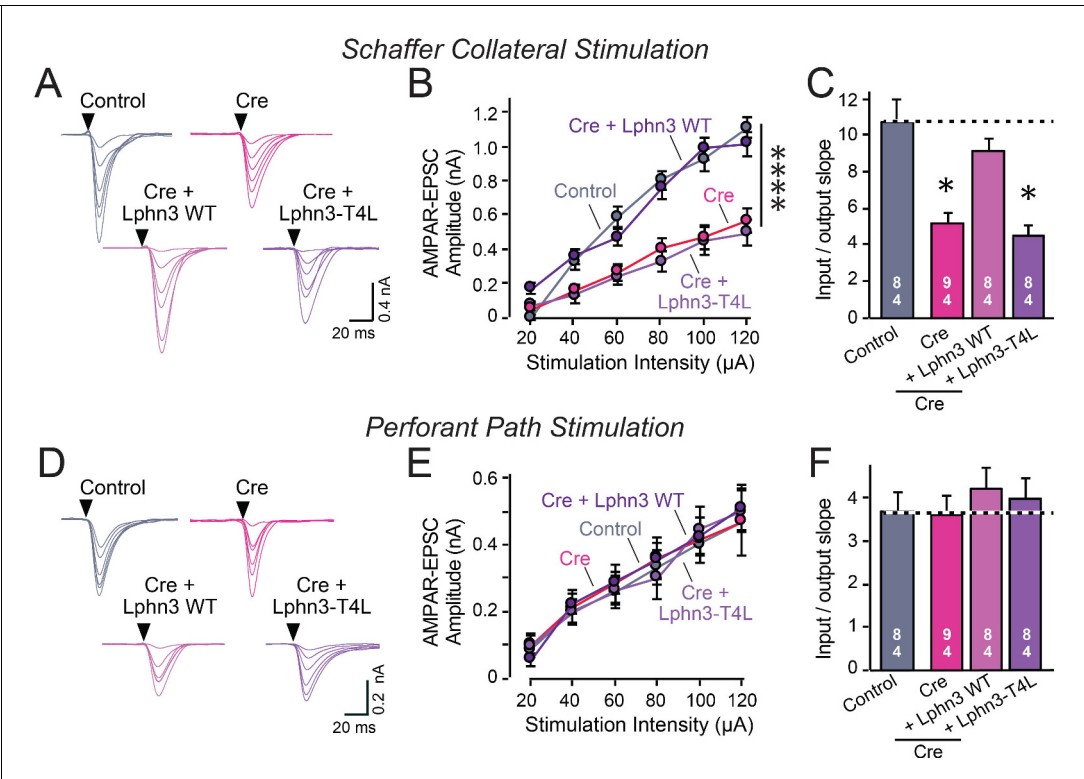

**Figure 5.** Abolishing Lphn3 GPCR activity by mutation of its third cytoplasmic loop blocks rescue of Schaffer collateral synaptic connectivity in CA1 region neurons in vivo. Data are from patch-clamp whole-cell recordings in acute hippocampal slice from Lphn3 conditional KO mice at P21-25. The CA1 region of the mice was injected at P0 with low titers of lentiviruses expressing either Cre alone, or Cre together with the indicated rescue constructs to induce sparse infection of CA1 pyramidal neurons (*Sando et al., 2019*). Control recordings were from uninfected cells in slices in the opposite uninfected hemisphere. (A–C) Sparse conditional KO of Lphn3 in CA1-region neurons impairs synaptic responses mediated by Schaffer-collateral inputs in a manner that can be rescued by wild-type but not T4L-mutant Lphn3 (A, sample traces; B, input-output curves; C, summary graph of the slopes of the input-output curves). (D–F) Sparse conditional KO of Lphn3 in CA1-region neurons has no effect on the synaptic responses mediated by perforant pathway inputs, and overexpression of wild-type or T4L-mutant Lphn3 does not alter these responses (D, sample traces; E, input-output curves; F, summary graph of the slopes of the input-output curves). Data are means ± SEM (numbers of cells/mice are indicated in bars). Statistical significance was assessed by one-way (For **C** and **F**) or two-way ANOVA with post hoc Tukey tests for multiple comparisons (For **B** and **E**) (**** denotes p<0.0001; * denotes p<0.05). See *Figure 4—figure supplement 1* for cellular capacitance and membrane resistance measurements. The online version of this article includes the following source data for figure 5:

**Source data 1.** Abolishing Lphn3 GPCR activity by mutation of its third cytoplasmic loop blocks rescue of Schaffer collateral synaptic connectivity in CA1 region neurons in vivo.

formation induced by conditional KO of Lphn3. Lphn2 was not tested in these experiments because they are labor-intensive and it was more relevant to examine its function in vivo. Third, analysis of the in vivo activity of WT and mutant Lphn2 and Lphn3 using sparse postsynaptic deletions of these molecules in hippocampal CA1 neurons in conditional KO mice demonstrated that WT Lphn2 and Lphn3 restored only the loss of synaptic connectivity of their cognate synapses, but had no effect on those synapses for which these latrophilins normally are not required. More importantly, these experiments established that mutant Lphn2 and Lphn3 that is unable to mediate GPCR signaling is incompetent to rescue the conditional KO phenotype. Together, these lines of evidence show that Lphn2 and Lphn3 function in synapse formation as GPCRs.

However, the present study has inherent limitations. Most importantly, our results do not identify the nature of the GPCR signal that is activated by latrophilins in synapse formation. Latrophilins have been reported to increase or decrease cAMP levels in cells in a constitutive manner (*Nazarko et al., 2018*; *Müller et al., 2015*; *Ovando-Zambrano et al., 2019*; *Scholz et al., 2017*). Our results support a function in increasing cAMP (*Figure 2*), but the physiological signaling activities of GPCRs are complex and what the precise type of signaling of latrophilins is in vivo may depend on the specific

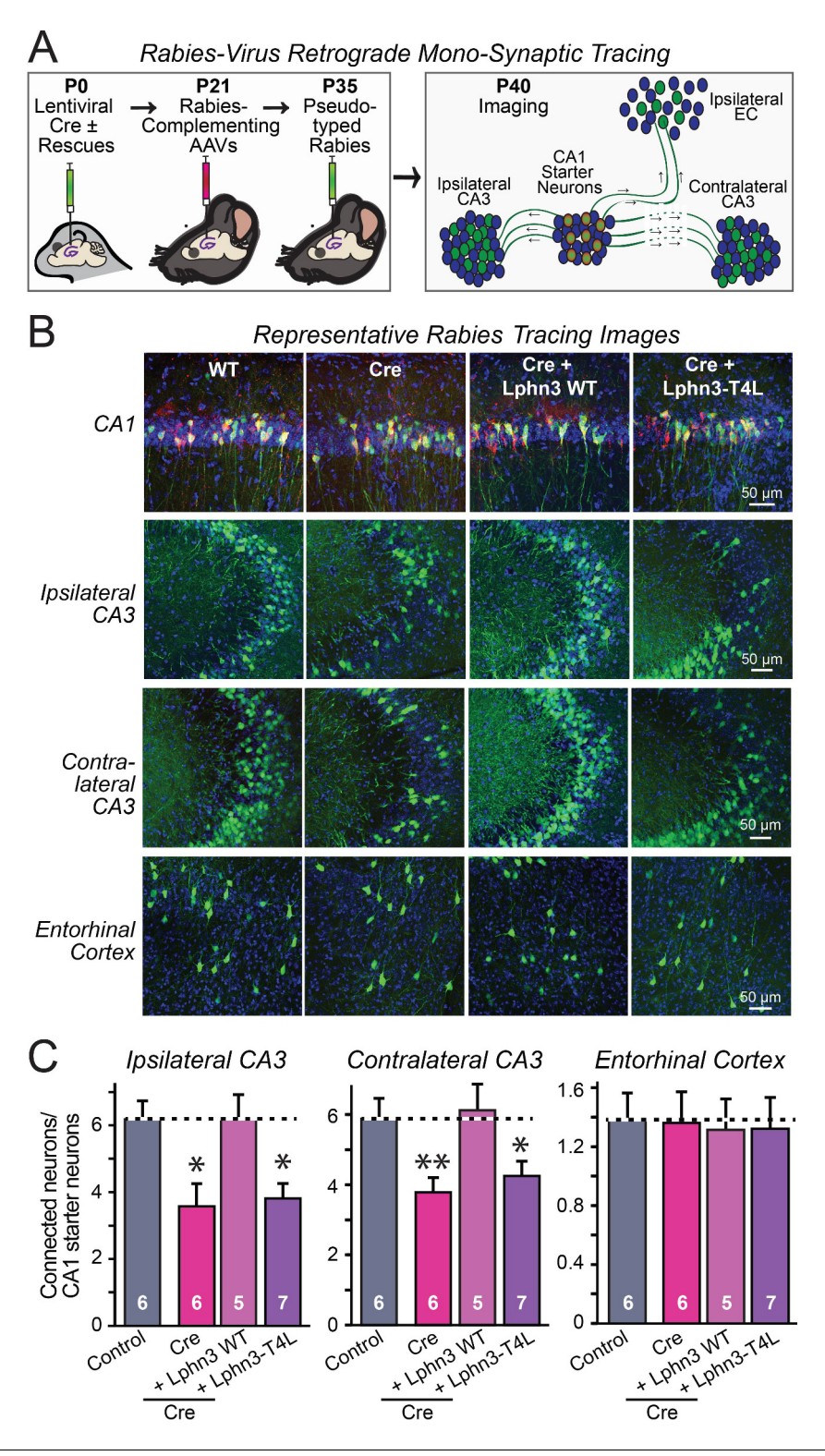

**Figure 6.** Retrograde mono-synaptic rabies virus tracing confirms that Lphn3 GPCR activity is essential for Schaffer-collateral synaptic connectivity in vivo. (**A**) Schematic of the experimental approach. The CA1 region of the hippocampus of Lphn3 conditional KO or control mice was injected at P0 with lentiviruses encoding Cre without or with the indicated rescue constructs, at P21 with AAVs encoding Cre-dependent AAVs of rabies-complementing proteins, and at P35 with pseudotyped rabies virus. Mice were analyzed by imaging at P40. (**B**)

*Figure 6 continued on next page*

*Figure 6 continued*

Representative images of monosynaptic rabies-virus tracing experiments from the four conditions analyzed. (**C**) Synaptic connectivity quantifications for the indicated synaptic inputs to CA1 region pyramidal neurons starter cells in the hippocampal CA1 region. Rabies virus tracing demonstrates that the impairment of Schaffer-collateral synaptic connectivity induced by deletion of Lphn3 is rescued by wild-type Lphn3 but not by T4L-mutant Lphn3. Data are means ± SEM (numbers of mice are indicated in bars). Statistical significance was assessed by one-way ANOVA with post hoc Tukey tests for multiple comparisons (For **C**) (** denotes p<0.01; * denotes p<0.05). The online version of this article includes the following source data for figure 6:

**Source data 1.** Retrograde mono-synaptic rabies virus tracing confirms that Lphn3 GPCR activity is essential for Schaffer-collateral synaptic connectivity in vivo.

context. Although our results establish that latrophilins act as GPCRs in synapse formation, they do not reveal what type of GPCR signaling drives synapse formation. The notion that this signaling may consist of increases in cAMP is attractive since pharmacological blockage of cAMP signaling impairs induction of spine synapses by glutamate (*Kwon and Sabatini, 2011*), but no other evidence currently shows that cAMP signaling – or any other GPCR signaling – mediates synapse formation. Thus, this question remains a challenge for future studies that will require new tools and approaches.

Moreover, the role of the cytoplasmic sequences of latrophilins remains unclear. These long sequences contain multiple highly conserved regions, including a C-terminal PDZ-domain binding motif (*Sugita et al., 1998*). What the function of these sequences is remains uncharacterized, although it is tempting to speculate that these sequences confer context specificity to the general GPCR activity of latrophilins by binding to specific interacting partners. Such partners remain to be identified, although SHANKs are prime candidates since they are postsynaptic scaffolding proteins that bind to latrophilins (*Kreienkamp et al., 2000*; *Tobaben et al., 2000*).

In summary, we here establish that Lphn2 and Lphn3 function as GPCRs in synapse formation, suggesting the possibility that compartmentalization of a canonical signaling pathway widely used in biology and controlled by GPCRs may be instrumental in driving the assembly of synaptic connections into circuits and control their specificity.

# Materials and methods

**Key resources table**

| Reagent type (species) or resource | Designation | Source or reference | Identifiers | Additional information |
|---|---|---|---|---|
| Genetic reagent (*Mus musculus*) | Lphn2 cKO | PMID:28972101 | JAX ID 023401 | |
| Genetic reagent (*Mus musculus*) | HA-Lphn3 cKO | PMID:30792275 | JAX ID 026684 | |
| Genetic reagent (*Mus musculus*) | CD-1 | Charles River Laboratory | | Pure CD-1 background mice used to generate primary hippocampal cultures for surface expression experiments. |
| Cell line (*Homo sapiens*) | HEK293T | ATCC | CRL-11268 | Mycoplasma testing was not performed since cells were maintained for a maximum of 10 passages. |
| Cell line (*Homo sapiens*) | HEK293F | Thermo Fisher | R79007 | Mycoplasma testing was not performed since cells were maintained for a maximum of 10 passages. |
| Recombinant DNA reagent | Lentiviral Syn NLS-GFP-Δcre | PMID:21241895 | | Also used in PMID:30792275 |
| Recombinant DNA reagent | Lentiviral Syn NLS-GFP-CRE | PMID:21241895 | | Also used in PMID:30792275 |

*Continued on next page*

*Continued*

| Reagent type (species) or resource | Designation | Source or reference | Identifiers | Additional information |
|---|---|---|---|---|
| Recombinant DNA reagent | pCMV WT hLphn3 | PMID:30792275, 26235030 | | N-terminal IgK signal peptide followed by FLAG |
| Recombinant DNA reagent | pCMV hFLRT3 | PMID:30792275, 26235030 | | N-terminal preprotrypsin signal peptide followed by MYC |
| Recombinant DNA reagent | pcDNA hTen2 | PMID:30792275, 26235030 | | C-terminal FLAG; extracellular HA prior to transmembrane region |
| Recombinant DNA reagent | pCAG WT rLphn2 | PMID:28972101, 30792275 | | N-terminal preprotrypsin signal peptide followed by FLAG; intracellular mVenus |
| Recombinant DNA reagent | pCMV Lphn3-ECD | This paper | | hLphn3 residues 20–923 with an N-terminal NCAM1 signal peptide and a C-terminal NCAM1 GPI-anchoring sequence |
| Recombinant DNA reagent | pCMV Lphn3-ΔCt | This paper | | hLphn3 with a stop codon at residue 1117, causing deletion of the C-terminal tail (amino acids 1117–1447) |
| Recombinant DNA reagent | pCMV Lphn3-T4L | This paper | | hLphn3 with T4 lysozyme peptide inserted at Ile1039 in intracellular loop 3 |
| Recombinant DNA reagent | pCAG Lphn2-T4L | This paper | | rLphn2 with T4 lysozyme peptide inserted at Gly1045 in intracellular loop 3 |
| Recombinant DNA reagent | Lentiviral Syn WT Lphn3 | PMID:30792275 | | Lentiviral shuttle plasmid containing FLAG-tagged hLphn3 |
| Recombinant DNA reagent | Lentiviral Syn Lphn3-ECD | This paper | | Lentiviral shuttle plasmid containing Lphn3-ECD for rescue experiments in primary cultures |
| Recombinant DNA reagent | Lentiviral Syn Lphn3-ΔCt | This paper | | Lentiviral shuttle plasmid containing Lphn3-ΔCt for rescue experiments in primary cultures |
| Recombinant DNA reagent | Lentiviral Syn Lphn3-T4L | This paper | | Lentiviral shuttle plasmid containing Lphn3-T4L for rescue experiments in primary cultures |
| Recombinant DNA reagent | Lentiviral Syn NLS-GFP-CRE p2a WT Lphn3 | PMID:30792275 | | Lentiviral shuttle plasmid co-expressing NLS-GFP-Cre and WT hLphn3 for in vivo rescue experiments |
| Recombinant DNA reagent | Lentiviral Syn NLS-GFP-CRE p2a Lphn3-T4L | This paper | | Lentiviral shuttle plasmid co-expressing NLS-GFP-Cre and hLphn3-T4L for in vivo rescue experiments |
| Recombinant DNA reagent | Lentiviral Syn NLS-GFP-CRE p2a WT Lphn2 | PMID:30792275 | | Lentiviral shuttle plasmid co-expressing NLS-GFP-Cre and WT rLphn2 for in vivo rescue experiments |

*Continued on next page*

*Continued*

| Reagent type (species) or resource | Designation | Source or reference | Identifiers | Additional information |
|---|---|---|---|---|
| Recombinant DNA reagent | Lentiviral Syn NLS-GFP-CRE p2a Lphn2-T4L | This paper | | Lentiviral shuttle plasmid co-expressing NLS-GFP-Cre and rLphn2-T4L for in vivo rescue experiments |
| Recombinant DNA reagent | Lentiviral EF1a phospho-PKA tdTomato | This paper | | A PKA phosphopeptide substrate (MRKVSKGEGGRRVSN) was fused to the N-terminus of tdTomato separated by a linker (GGSGGGSGG) and was detected with the PKA phosphopeptide substrate antibody (sc-56941) |
| Recombinant DNA reagent | pink Flamindo cAMP reporter | PMID:28779099; Addgene | RRID:addgene_102356 | |
| Recombinant DNA reagent | AAV CAG FLEX TVA-mCherry | PMID:26232228; Addgene | RRID:addgene_48332 | Rabies complementing AAV shuttle plasmid |
| Recombinant DNA reagent | AAV CAG FLEX rG | PMID:26232228; Addgene | RRID:addgene_48333 | Rabies complementing AAV shuttle plasmid |
| Antibody | Anti-HA mouse monoclonal | Covance | #MMS101R | 1:500 IHC; 1:1,000 ICC; 1:2,000 IB |
| Antibody | Anti-HA rabbit monoclonal | Cell Signaling Technologies | #3724 | 1:2,000 ICC; 1:1,000 IHC |
| Antibody | Anti-FLAG mouse monclonal | Sigma | #F3165 | 1:2,000 IB |
| Antibody | Anti-beta Actin mouse monoclonal | Sigma | #A1978 | 1:10,000 IB |
| Antibody | Anti-MAP2 chicken polyclonal | Encor | #CPCA MAP2 | 1:2,000 ICC |
| Antibody | Anti-PSD95 mouse monoclonal | Synaptic Systems | #124011 | 1:2,000 ICC |
| Antibody | Anti-vGLUT1 guinea pig polyclonal | Millipore | #AB5905 | 1:2,000 ICC |
| Antibody | Anti-Homer1 rabbit polyclonal | Synaptic Systems | #160003 | 1:2,000 ICC |
| Antibody | Anti-PKA phosphopeptide substrate mouse monoclonal | Santa Cruz Biotechnology | #sc-56941 | 1:1,000 ICC |
| Chemical compound, drug | Papain | Worthington | #LS003127 | |
| Chemical compound, drug | Matrigel | Corning | #356235 | |
| Chemical compound, drug | B-27 supplement | Gibco | #17504044 | |
| Chemical compound, drug | Cytosine arabinofuranoside | Sigma | #C6645 | |
| Chemical compound, drug | DAPI | Roche | #10236276001 | |

*Continued on next page*

*Continued*

| Reagent type (species) or resource | Designation | Source or reference | Identifiers | Additional information |
|---|---|---|---|---|
| Chemical compound, drug | QX-314 | Tocris | #1014 | |
| Chemical compound, drug | Picrotoxin | Tocris | #1128 | |
| Chemical compound, drug | Tetrodotoxin | Tocris | #1069 | |
| Chemical compound, drug | Freestyle MAX reagent | Life Technologies | #16447100 | |
| Software, algorithm | SnapGene | GSL Biotech | | previously existing |
| Software, algorithm | pClamp10 | Molecular Devices | | previously existing |
| Software, algorithm | Clampfit10 | Molecular Devices | | previously existing |
| Software, algorithm | NIS-Elements AR | Nikon | | previously existing |
| Software, algorithm | ImageJ | National Institutes of Health | | previously existing |
| Software, algorithm | Adobe Photoshop | Adobe | | previously existing |
| Software, algorithm | Adobe Illustrator | Adobe | | previously existing |
| Software, algorithm | Graphad Prism 8.0 | Graphad software | | previously existing |

*For antibody dilutions: IHC – immunohistochemistry; ICC – immunocytochemistry; IB – immunoblotting.

## Mice

HA-Lphn3 cKO and Lphn2 cKO mice were previously described[12,13]. Overexpression studies were conducted in primary hippocampal cultures from CD1 mice. Mice were weaned at 18–21 days of age and housed in groups of 2 to 5 on a 12 hr light/dark cycle with food and water *ad libidum*. Stanford Animal Housing Facility: All procedures conformed to National Institutes of Health Guidelines for the Care and Use of Laboratory Mice and were approved by the Stanford University Administrative Panel on Laboratory Animal Care.

## Plasmids

Schematic designs of all constructs utilized in this study are shown in Extended Data *Figure 1*. Lentiviruses expressing NLS-GFP-ΔCre or NLS-GFP-Cre driven by the synapsin-1 promoter were described previously (*Kaeser et al., 2011*) and were used for all Lphn3 cKO experiments in cultured neurons or in in vivo manipulations. Lphn2 expression plasmids were based on rat Lphn2 lacking SSA, with an exogenous N-terminal signal peptide from preprotrypsin followed by a FLAG epitope (sequence: **MSALLILALVGAAVA**DYKDDDDKLFSRAALPFGLVRRE...; signal peptide bolded and FLAG epitope underlined), while the Lphn3 expression plasmids were based on human Lphn3 containing an insert in SSA, with an exogenous N-terminal signal peptide from IgKappa followed by a FLAG epitope (sequence: METDTLLLWVLLLWVPGSTGDAGAQDYKDDDDKFSRAPIPMAVVRRE...).

All latrophilin expression experiments in HEK293T cells were conducted with pCMV5 or pcDNA3 vectors using the CMV or CAG promoter, and all experiments in neurons and in vivo with lentiviruses

in which latrophilin expression was driven by the rat synapsin-1 promoter. For Lphn2 and Lphn3 rescue experiments, the following mutations and truncations were generated:

### Lphn3-ECD

Encodes the Lphn3 extracellular sequence (residues 20–923) with an N-terminal NCAM1 signal peptide and the C-terminal NCAM1 GPI-anchoring sequence.

### Lphn2-T4L and Lphn3-T4L

Encodes Lphn2 or Lphn3 with an insertion of the T4 lysozyme sequence into the intracellular loop three at Gly1045 of Lphn2 or at Ile1039 of Lphn3 (T4 lysozyme sequence: LENIFEMLRIDEGLRLKIYKD TEGYYTIGIGHLLTKSPSLNAAKSELDKAIGRNTNGVITKDAEKLFNQDVDAAVRGILRNAKLKPVYDSLDA VRRAALINMVFQMGETGVAGFTNSLRMLQQKRWDEAAVNLAKSRWYNQTPNRAKRVITTFRTGTWDA Y).

### Lphn3-ΔCt

Encodes Lphn3 with a stop codon at residue 1117, causing deletion of the C-terminal tail (amino acids 1117–1447). VRKEYGKCLR* For phosphodiesterase PDE7b expression experiments, human PDE7b cDNA was used, with HEK293T cell expression mediated by pCMV5 or pcDNA3 vectors.

### Phospho-PKA tdTomato reporter

A PKA phosphopeptide substrate (MRKVSKGEGGRRVSN) was fused to the N-terminus of tdTomato, separated by a linker (GGSGGGSGG) and was detected via ICC with the PKA Phosphopeptide Substrate Antibody (Santa Cruz Biotechnology, BDI251, Cat# sc-56941; 1:1,000 ICC).

Pink Flamindo cAMP Reporter: This plasmid was a gift from Tetsuya Kitaguchi (Addgene plasmid #102356).

Rabies complementing AAVs: AAV CAG FLEX TVA-mCherry and AAV CAG FLEX RG were gifts from Liqun Luo (Addgene #48332 and #48333).

### Cell lines

HEK293T from ATCC (CRL-11268). HEK293F cells from ThermoFisher (R79007). Mycoplasma testing was not routinely performed since cells were only passaged for a maximum of 10 passages before utilizing a fresh aliquot from the supplier. Cells were passaged at 90% confluency (for HEK293T cells) or at 4 million cells/mL (for HEK293F cells) and counted using Trypan blue to assess viability. Unhealthy cultures (viability <90%) were discarded and replaced with fresh aliquots of the respective cell lines.

### Antibodies

The following antibodies were used at the indicated concentrations (IHC-immunohistochemistry; ICC-immunocytochemistry; IB-immunoblot): anti-HA mouse (Covance Cat# MMS101R; 1:1,000 ICC, 1:2,000 IB), anti-HA rabbit (Cell Signaling Technologies Cat# 3724; 1:2,000 ICC), anti-FLAG mouse (Sigma Cat#F3165; 1:2,000 IB), anti-βactin mouse (Sigma Cat#A1978; 1:10,000 IB), anti-Map2 chicken (Encor Cat# CPCA MAP2; 1:2,000 ICC), anti-PSD95 mouse (Synaptic Systems Cat# 124011; 1:2,000 ICC), anti-vGLUT1 guinea pig (Millipore Cat# AB5905; 1:2,000 ICC), anti-Homer1 rabbit (Synaptic Systems Cat#160003; 1:2,000 ICC); PKA Phosphopeptide Substrate Antibody (Santa Cruz Biotechnology, BDI251, Cat# sc-56941; 1:1,000 ICC), fluorescently-conjugated goat secondary antibodies from Life Technologies.

### Cultured hippocampal neurons

Hippocampi were dissected from P0 Lphn3fl/fl or CD-1 mice, and cells were dissociated by papain (Worthington, Cat# LS003127) digestion for 20 min at 37°C, filtered through a 70 µm cell strainer (Falcon Cat# 352350), and plated on Matrigel (Corning Cat# 356235)-coated 0 thickness glass coverslips (Assistent Cat# 01105209) in 24-well plates. Plating media contained 5% fetal bovine serum (Atlanta), B27 (Gibco Cat# 17504044), 0.4% glucose (Sigma), 2 mM glutamine (Gibco Cat# 25030164), in 1x MEM (Gibco Cat# 51200038). Culture media was exchanged to Growth media 24 hr later (1 DIV), which contained 5% fetal bovine serum (Atlanta), B27 (Gibco), 2 mM glutamine

(Gibco) in Neurobasal A (Gibco Cat# 10888022).Cytosine β-D-arabinofuranoside hydrochloride (Sigma Cat#C6645) was added at a final concentration of 4 µM on 3 DIV in a 50% growth media exchange, and neurons were analyzed 14–16 DIV.

## Virus production

For production of lentiviruses, the lentiviral expression shuttle vector and three helper plasmids (pRSV-REV, pMDLg/pRRE and vesicular stomatitis virus G protein (VSVG)) were co-transfected into HEK293T cells (ATCC), at 5 µg of each plasmid per 25 cm$^2$ culture area, respectively. Control conditions aside from GFP-ΔCre or GFP-infected controls were lentivirus produced with empty shuttle vector. Transfections were performed using the calcium-phosphate method. Media with viruses was collected at 48 hr after transfection, centrifuged at 5000 x g for 5 min to pellet cellular debris, filtered (0.45 µm pore size), and added directly to primary cultures for in vitro experiments. For in vivo injections, virus was produced at the Janelia Farm Virus Core Facility where infectious titer was determined, and equal numbers of viral particles were injected for each experiment.

## Immunocytochemistry

All solutions were made fresh and filtered via a 0.2 µm filter prior to starting experiments. Cells were washed briefly 1 x with PBS, fixed with 4% PFA (Electron Microscopy Science Cat# 15714)/4% sucrose/PBS for 20 min at 4°C, washed 3 × 5 min in PBS, and permeabilized in 0.2% Triton x100/PBS for 5 min at room temperature. Cells were subsequently placed in blocking buffer (4% BSA (Sigma Cat# 10735086001)/3% goat serum(Sigma Cat# G9023)/PBS) 1 hr, incubated with diluted primary antibodies in blocking buffer overnight at 4°C, washed 3 × 5 min in PBS, incubated with diluted fluorescently-conjugated secondary antibodies in blocking buffer for 1 hr, counterstained with DAPI in PBS for 15 mins at room temperature (Sigma Cat# 10236276001), washed 3 x in PBS, and mounted on UltraClear microscope slides (Denville Scientific Cat# M1021) using Fluoromount-G (Southern Biotech Cat#0100–01). For surface labeling of FLAG-tagged Lphn constructs, cultures were immunolabeled for surface FLAG by staining without permeabilization, and subsequently permeabilized in 0.2% Triton X-100/PBS and immunolabeled for MAP2.

## Immunoblotting

Samples were collected in sample buffer containing 312.5 mM Tris-HCl pH 6.8, 10% SDS, 50% glycerol, 12.5% 2-mercaptoethanol, bromophenol blue, and protease inhibitors (Sigma Cat#5056489001) and run on SDS-PAGE gels (8% PAGE gels for FLAG-Lphn immunoblots and 12% PAGE gels β-actin immunoblots) at 30 mA/gel constant current. The Precision Plus Protein Dual Color Protein Standard (BioRad Cat# 1610374) was used as a protein molecular weight ladder. Protein was transferred onto nitrocellulose transfer membrane in transfer buffer (25.1 mM Tris, 192 mM glycine, 20% methanol) at 200 mA constant current for 2 hr at 4°C. Membranes were blocked in 5% non-fat dry milk/TBST (20 mM Tris-HCl pH 7.4, 150 mM NaCl, 0.05% Tween-20) for 1 hr at room temperature, incubated in primary antibodies diluted into 5% milk/TBST overnight at 4°C, washed 3 × 5 min in TBST, incubated in corresponding secondary antibodies (Licor IRDye 800CW donkey anti-mouse Cat#92632212; IRDye 800CW Donkey anti-rabbit Cat#92632213) diluted into TBST, washed 5 × 5 min in TBST, and imaged on a Licor Odyssey system.

## Electrophysiology of cultured neurons

For whole-cell patch clamp physiology experiments, the patch pipettes were pulled from borosilicate glass capillary tubes (World Precision Instruments, Cat# TW150-4) using a PC-10 pipette puller (Narishige). The resistance of pipettes filled with intracellular solution varied between 2–4 MOhm. Synaptic currents were monitored with a Multiclamp 700B amplifier (Molecular Devices). The frequency, duration, and magnitude of the extracellular stimulation were controlled with a Model 2100 Isolated Pulse Stimulator (A-M Systems, Inc) synchronized with Clampex 10 data acquisition software (Molecular Devices). For excitatory voltage-clamp recordings, a whole-cell pipette solution was used containing (in mM) 135 Cs-Methanesulfonate, 8 CsCl, 10 HEPES, 0.25 EGTA, 0.3 Na$_2$GTP, 2 MgATP, seven phosphocreatine, and 10 QX-314 (Tocris Cat#1014) (pH 7.3, adjusted with CsOH and 303 Osm). The external bath solution contained (in mM) 140 NaCl, 5 KCl, 2 CaCl$_2$, 0.8 MgCl$_2$, 10 HEPES, 10 glucose (pH 7.4, adjusted with NaOH). AMPAR- and NMDAR-EPSC postsynaptic currents were

pharmacologically isolated by adding the GABA$_A$ receptor blocker picrotoxin (50 μM) (Tocris Cat#1128) to the extracellular bath solution. AMPAR-EPSCs and mEPSC recordings were performed while holding the cell at −70 mV, and NMDAR-EPSCs at +40 mV. Spontaneous miniature excitatory postsynaptic currents (mEPSCs) were monitored in the presence of tetrodotoxin (1 μM) (Tocris Cat#1069) to block action potentials, at −70 mV holding potential. Synaptic currents were sampled at 10 kHz and analyzed offline using Clampfit 10 (Molecular Devices) software. Miniature events were analyzed using the template matching search and a minimal threshold of 5 pA and each event was visually inspected for inclusion or rejection by an experimenter blind to the recording condition.

## Slice electrophysiology

For acute slice electrophysiology, lentiviruses were injected into P0 mice, and infected CA1 pyramidal neurons were analyzed at P21-25. Transverse hippocampal slices (300 μm) were prepared by cutting in ice-cold solution containing (in mM): 228 Sucrose, 2.5 KCl, 1 NaH$_2$PO$_4$, 26 NaHCO$_3$, 0.5 CaCl$_2$, 7 MgSO$_4$, 11 D-Glucose saturated with 95% O$_2$/5% CO$_2$. Slices were transferred to a holding chamber containing artificial cerebrospinal fluid (ACSF, in mM): 119 NaCl, 2.5 KCl, 1 NaH$_2$PO$_4$, 26 NaHCO$_3$, 2.5 CaCl$_2$, 1.3 MgSO$_4$, 11 D-Glucose, ~290 mOsm. Slices recovered at 32°C for 30 min, followed by holding at room temperature for 1 hr. Acute slices were transferred to a recording chamber continuously superfused with oxygenated ACSF (1.5 ml/min) maintained at 32°C. Neurons were clamped at −70 mV, and two-pathways of extracellular-evoked EPSCs in hippocampal slices were monitored. AMPAR-EPSCs were evoked by electrical stimulation using tungsten electrodes (Matrix electrode, 2 × 1, FHC Cat# MX21AEW(RT2)) positioned at the S. radiatum 150 μm proximal to CA3, and the S. lacunosum-moleculare proximal to the molecular layer of the dentate gyrus.

## Sparse lentiviral infections in vivo

P0 Lphn2 fl/fl orLphn3 fl/fl pups were anesthetized on ice for 5 min and subsequently placed in a homemade clay mold. Lentivirus was loaded and injected with a glass pipette connected to an infusion pump (Harvard Apparatus) completely continuous with mineral oil. A stereotactic injection rig (Kopf) was used to target injections. Lentiviruses were diluted to a 1 × 10$^7$ IU/mL titer and 0.3 μL were injected at the following stereotactic coordinates from the lambda at a 1 μL/min rate: A-P 1.0, M-L 0.9, D-V three subsequent injections at 1.4, 1.2, 1.0. Pups then recovered in a clean cage placed on a heating pad, and were transferred back to their home cage after completely recovering.

## Monosynaptic retrograde rabies tracing

Lentiviral injections were performed at P0 as described above. Complementing AAVs containing CAG-FLEX-TCB-mCherry and CAG-FLEX-RG were generated at the Janelia Farm Viral Core Facility in capsid 2/5 and 2/8, respectively, and injected in P21 mice. Adult mice were stereotactically injected mice by anesthetizing with an intraperitoneal injection of tribromoethanol (Avertin) (250 mg/kg). Mouse heads were shaved, the shaved area was cleaned with Betadine, lubricant was placed on eyes (Puralube Vet Ointment), and heads were secured in a stereotactic injection rig (Kopf) and a small incision was made through the scalp with sterilized tools. Viral solution was injected with a glass pipette at a flow rate of 0.15 μL/min and 0.5 μL volume per injection. Coordinates used for unilateral CA1 injections were AP −1.80 mm, ML +/− 1.35 mm, DV −1.25 mm. The injection pipette was left at the injection site for five mins after the injection to prevent the spread of virus into neighboring brain regions. Incisions were sutured and sealed with Vetbond tissue adhesive (1469 SB), and mice were subsequently removed from the stereotactic injection rig. Mice were monitored in a warmed recovery cage until full recovery. RbV-CVS-N2c-deltaG-GFP (EnvA) was produced at the Janelia Farm Viral core facility and injected 2 weeks after AAV injections at an infectious titer of 1 × 10$^8$ IU/mL and 0.5 μL volume as described above. Mice were subsequently perfused and analyzed 5 days later. Brains were post-fixed overnight in 4% PFA/PBS and sliced on a vibratome in 50 μm sections. Sections were labeled with DAPI, washed with PBS, mounted and imaged using a Nikon A1 confocal microscope with a 20x objective.

## Imaging

Images were acquired using a Nikon A1 Eclipse Ti confocal microscope with a 10x, 20x, and 60x objective, operated by NIS-Elements AR v4.5 acquisition software. Laser intensities and acquisition

settings were established for individual channels and applied to entire experiments. Image analysis was conducted using Nikon Elements, ImageJ, and Adobe Photoshop.

### In vitro synapse imaging

Primary hippocampal cultures infected with lentiviruses encoding GFP-ΔCre or GFP-Cre with or without co-expression of indicated rescue cDNAs were immunolabeled for vGluT1 (guinea pig, Millipore Cat# AB5905; 1:2,000 ICC; in 488 channel), PSD-95 (mouse, Synaptic Systems Cat# 124011; 1:2,000 ICC; in 546 channel), and MAP2 (chicken, Encor Cat# CPCA MAP2; 1:2,000 ICC; in 647 channel). Images were collected as described above with a 60x objective at 0.1 μm/pixel resolution and 0.2 μm z-stack intervals. Overlapping vGLUT1/PSD-95 puncta were automatically quantified along measured distances of MAP2-labeled primary and secondary dendrites using Nikon NIS-Elements AR software.

### Cell aggregation assay

FreeStyle HEK293F cells (Life Technologies Cat# R79007) grown to a density of $1 \times 10^6$ cells/mL in a 30 mL volume were co-transfected with 30 μg of either pCMV5-Emerald or pCMV5-tdTomato and 30 μg of the indicated construct using a FreeStyle Max reagent (Life Technologies Cat# 16447100). All cDNAs were encoded in the pCMV5 vector. FreeStyle HEK293F cells were grown at 37°C/8%$CO_2$ with shaking at 125 rpms. Transfected cells were mixed in non-coated 12-well plates at a 1:1 ratio 2-days post-transfection and subsequently incubated for an additional 2 hr. Live cells were imaged in 12-well plates using a Nikon A1 confocal microscope. Aggregation index was calculated using ImageJ, measuring the percentage of signal/frame occupied by cells forming complexes of two or more cells relative to the total signal of the frame.

### Statistics

Experiments were performed in a blinded manner whenever possible by coding viral solutions. All data are expressed as means ± SEM and represent the results of at least three independent biological replicates. Statistical significance was determined using the two-tailed Student's t-test, one-way ANOVA, or two-way ANOVA with post hoc Tukey tests for multiple comparisons, as indicated in the Figure Legends. Data analysis and statistics were performed with Microsoft Excel and GraphPad Prism 8.0. All numerical data and statistical parameters within this study can be located in the Source Data file.

## Acknowledgements

We thank Irina Huryeva for technical assistance, and Shoji Maeda (Kobilka laboratory, Stanford University), Brian Kobilka (Stanford University) and Demet Arac (University of Chicago) for extensive discussions and advice.

## Additional information

### Funding

| Funder | Grant reference number | Author |
| --- | --- | --- |
| National Institute of Mental Health | K99-MH117235 | Richard Sando |

The funders had no role in study design, data collection and interpretation, or the decision to submit the work for publication.

### Author contributions

Richard Sando, Conceptualization, Resources, Data curation, Formal analysis, Supervision, Funding acquisition, Validation, Investigation, Visualization, Methodology, Writing - original draft, Project administration, Writing - review and editing; Thomas C Südhof, Conceptualization, Resources, Data curation, Supervision, Funding acquisition, Validation, Investigation, Visualization, Writing - original draft, Project administration, Writing - review and editing

**Author ORCIDs**
Richard Sando (iD) https://orcid.org/0000-0002-1797-2346
Thomas C Südhof (iD) https://orcid.org/0000-0003-3361-9275

## Ethics

Animal experimentation: All procedures strictly conformed to National Institutes of Health Guidelines for the Care and Use of Laboratory Mice and were approved by the Stanford University Administrative Panel on Laboratory Animal Care (APLAC) and institutional animal care and use committee (IACUC). The animal protocol #20787 was approved by Stanford University APLAC and IACUC. All surgeries were performed under Avertin anesthesia and buprenorphine analgesia, and every effort was made to minimize suffering, pain and distress.

## Decision letter and Author response

Decision letter https://doi.org/10.7554/eLife.65717.sa1
Author response https://doi.org/10.7554/eLife.65717.sa2

## Additional files

### Supplementary files
• Transparent reporting form

### Data availability

All raw numerical data within the study has been submitted together with the manuscript.

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
