## [Decision Letter]

**Acceptance summary:**

The main finding that GPCR activity is necessary for latrophilins' role in synapse formation is both surprising and important. This work will inspire new research on compartmentalized GPCR signaling at the synapse.

**Decision letter after peer review:**

Thank you for submitting your article "Latrophilin GPCR Signaling Mediates Synapse Formation" for consideration by *eLife*. Your article has been reviewed by three peer reviewers, including Graeme W Davis as the Reviewing Editor and Reviewer #1, and the evaluation has been overseen by Gary Westbrook as the Senior Editor. The reviewers have discussed their reviews with one another, and the Reviewing Editor has drafted this to help you prepare a revised submission.

Essential Revisions:

1) The manuscript was viewed as largely ready for acceptance “as is”. There are two related points, however, raised by two of the reviewers, that should be addressed. The first point concerns the “trafficking” of the mutant proteins to spines. There was discussion among the reviewers on this point and it was agreed that “trafficking” per se is a loaded term that might justify additional experimentation to quantify intracellular versus extracellular tagged proteins, among other variables. However, this would be tangential to the main conclusions of the work. The reviewers suggest a rephrasing of the text, referring to experimental “controls” for the presence of the transgenic proteins at spines, or some similar modification.

2) The reviewers thought that additional verification of the cellular assay for cell permeability in PFA treated samples was necessary. Is the PFA treated sample truly impermeable? This control could be performed easily. Alternatively, if the protocol has been routinely used and verified, this could be stated as such, though we think it in the interest of the authors to provide a quick experimental control here. One reviewer writes, "A key argument is that only surface labeled latrophilin is visualized, but it is possible that 4% PFA itself permeabilizes neurons to some extent and some intracellular antigens may be stained. The methods sound as though surface labeling was done after PFA fixation, but uncertainty remains whether cells were fixed or not before surface labeling. Finally, the quantification of the signal itself is not described at all (in the Materials and methods or main text), and it is difficult to assess what the ratio of surface flag/Map2 signal means. The authors should clarify these concerns by better describing what was done and how the labeling was quantified. Additional controls for the surface labeling (for example that non-surface antigens in the same compartment – PSD-95 comes to mind – are not labeled under these conditions)"

3) Some additional text corrections are noted:

– Introduction: Wang et al., 2020, is not listed in the reference list.

– Subsection “Lphn2 and Lphn3 function as GPCRs in the hippocampus in vivo” and Figures 4/5: the expression system for rescue is not described at all in the main text and figures. Maps of the 2A-mediated bicistronic expression in the figure and explanations in the text would make the experiment more understandable.

– Subsection “Lphn2 and Lphn3 function as GPCRs in the hippocampus in vivo” and on: the logic and approach of the retrograde tracing can be patched together from the cited literature, Materials and methods and figures, but it would be very helpful to add a couple of sentences on how this experiment was done and quantified. The description in the main text is so short that it cannot be followed by a broader audience.

– Figure 1 legend and other figure legends: no statistics are listed for Figure 1C and D, but they should be performed and listed. In general, how the statistics were done is difficult to understand from the information that is provided, and more precision in figure legends etc is necessary. For example, Figures 4B, E and 5B, E list ANOVA testing and four stars****, but there is no explanation what p-value the four stars**** stand for, and it is unclear whether the ANOVA result (for both variables and interaction) or the post-test result is reported (and if so, why).

---

## [Author Response]

Essential Revisions:1) The manuscript was viewed as largely ready for acceptance “as is”. There are two related points, however, raised by two of the reviewers, that should be addressed. The first point concerns the “trafficking” of the mutant proteins to spines. There was discussion among the reviewers on this point and it was agreed that “trafficking” per se is a loaded term that might justify additional experimentation to quantify intracellular versus extracellular tagged proteins, among other variables. However, this would be tangential to the main conclusions of the work. The reviewers suggest a rephrasing of the text, referring to experimental “controls” for the presence of the transgenic proteins at spines, or some similar modification.

We agree that we cannot claim trafficking of mutant protein to dendritic spines with the present data. Also, definitively determining this would require extensive additional experimentation that would indeed be tangential to our main conclusions. These types of experiments would be further complicated by our previous observations (Sando, Jiang and Südhof, 2019) that different Lphn isoforms are present in non-overlapping populations of synapses even in neuronal cultures. Therefore, we rephrased the text to state that we detected the presence of mutant protein on the neuronal cell surface, rather than trafficking to dendritic spines. We feel this more accurately describes our data and still validates the utility of these mutants for our experimental purposes.

2) The reviewers thought that additional verification of the cellular assay for cell permeability in PFA treated samples was necessary. Is the PFA treated sample truly impermeable? This control could be performed easily. Alternatively, if the protocol has been routinely used and verified, this could be stated as such, though we think it in the interest of the authors to provide a quick experimental control here. One reviewer writes, "A key argument is that only surface labeled latrophilin is visualized, but it is possible that 4% PFA itself permeabilizes neurons to some extent and some intracellular antigens may be stained. The methods sound as though surface labeling was done after PFA fixation, but uncertainty remains whether cells were fixed or not before surface labeling. Finally, the quantification of the signal itself is not described at all (in the Materials and methods or main text), and it is difficult to assess what the ratio of surface flag/Map2 signal means. The authors should clarify these concerns by better describing what was done and how the labeling was quantified. Additional controls for the surface labeling (for example that non-surface antigens in the same compartment – PSD-95 comes to mind – are not labeled under these conditions)".

This is a very important point that required additional experimentation. Some fixation methods such as methanol fixation are well-known to also permeabilize the cell membrane. We performed this control experiment exactly as requested by the reviewer, except we immunolabeled for the intracellular excitatory synaptic scaffold protein Homer1 (the reviewer suggested PSD-95) and also included the positive control of FLAG-WT Lphn3 overexpression via lentivirus. Please see updated Figure 1—figure supplement 1B for new experimental data addressing this point. We subjected 12DIV primary hippocampal cultures to our fixation procedure (4% PFA/4% sucrose/PBS for 20 minutes at 4°C), washed 3x with PBS, and either treated cells with 0.2% Triton X-100/PBS for 5 minutes followed by blocking, or moved directly to blocking (non-permeabilized). We subsequently labeled for FLAG together with Homer1. We observed minor signal in the Homer1 channel in non-permeabilized conditions, although it is difficult to discern actual Homer1 signal vs. background. Permeabilization with 0.2% Triton X-100/PBS exposed a substantial amount of Homer1 signal, while FLAG-Lphn3 signal was indistinguishable between permeabilized and non-permeabilized conditions. Therefore, while we cannot exclude that our fixation procedure induces a minor degree of cell permeability, the vast majority of intracellular antigen is inaccessible in our fixation procedure alone. We edited the text to reflect this important point, and are grateful to the reviewers for suggesting this important experimental control.

3) Some additional text corrections are noted:– Introduction: Wang et al., 2020, is not listed in the reference list.

Thank you for identifying this oversight. We have added the reference to the reference list.

– Subsection “Lphn2 and Lphn3 function as GPCRs in the hippocampus in vivo” and Figures 4/5: the expression system for rescue is not described at all in the main text and figures. Maps of the 2A-mediated bicistronic expression in the figure and explanations in the text would make the experiment more understandable.

We have added maps of the 2A-mediated bicistronic vectors to Figure 4—figure supplement 1.

– Subsection “Lphn2 and Lphn3 function as GPCRs in the hippocampus in vivo” and on: the logic and approach of the retrograde tracing can be patched together from the cited literature, Materials and methods and figures, but it would be very helpful to add a couple of sentences on how this experiment was done and quantified. The description in the main text is so short that it cannot be followed by a broader audience.

We have added additional experimental details to the text (Results).

– Figure 1 legend and other figure legends: no statistics are listed for Figure 1C and D, but they should be performed and listed. In general, how the statistics were done is difficult to understand from the information that is provided, and more precision in figure legends etc is necessary. For example, Figures 4B, E and 5B, E list ANOVA testing and four stars****, but there is no explanation what p-value the four stars**** stand for, and it is unclear whether the ANOVA result (for both variables and interaction) or the post-test result is reported (and if so, why).

We have added additional descriptions of the statistical analyses in the figure legends. Furthermore, we have added all of the statistical parameters and analyses into the attached Source Data files, so now all raw quantitative data and statistics are provided.